# Combined Regularized Discriminant Analysis and Swarm Intelligence Techniques for Gait Recognition

**DOI:** 10.3390/s20236794

**Published:** 2020-11-27

**Authors:** Tomasz Krzeszowski, Krzysztof Wiktorowicz

**Affiliations:** Faculty of Electrical and Computer Engineering, Rzeszow University of Technology, al. Powstancow Warszawy 12, 35-959 Rzeszow, Poland; kwiktor@prz.edu.pl

**Keywords:** gait recognition, biometrics, regularized discriminant analysis, particle swarm optimization, grey wolf optimization, whale optimization algorithm

## Abstract

In the gait recognition problem, most studies are devoted to developing gait descriptors rather than introducing new classification methods. This paper proposes hybrid methods that combine regularized discriminant analysis (RDA) and swarm intelligence techniques for gait recognition. The purpose of this study is to develop strategies that will achieve better gait recognition results than those achieved by classical classification methods. In our approach, particle swarm optimization (PSO), grey wolf optimization (GWO), and whale optimization algorithm (WOA) are used. These techniques tune the observation weights and hyperparameters of the RDA method to minimize the objective function. The experiments conducted on the GPJATK dataset proved the validity of the proposed concept.

## 1. Introduction

Biometric authentication (also known as biometrics) refers to identifying or verifying individuals based on their biological or behavioral traits [1]. There are many different biometric traits among which can be distinguished the face, iris, fingerprint, palm print, voice, signature, or gait. Typically, gait is a manifestation of an individual’s walking style; hence, its recognition means identifying a person by his/her way of walking. The major advantages of gait are: noninvasive, can be captured at a distance, hard to conceal, and non-cooperative. These advantages make it an ideal trait for visual surveillance systems [2]. However, the recognition performance of existing methods is limited by the influence of a large number of covariate factors affecting both appearance and dynamics of the gait, e.g., variations in footwear and clothing, viewpoint variations, changes in the characteristics of the surface on which movement occurs, various carrying conditions, injuries affecting movement, and so on. These are the reasons why gait recognition has been extensively studied in recent years.

Gait recognition methods [3] can be categorized as model-free (appearance-based) [2,4,5,6,7,8,9,10,11,12] and model-based [13,14,15,16,17,18,19,20,21,22]. Most gait recognition studies are based on model-free approaches that employ the whole motion pattern of the human body. Several techniques were proposed to characterize this motion pattern, such as the gait energy image (GEI) [2,5,7,10], which is a spatio-temporal gait representation, GEI region bounded by legs (RBL) [8], human body contours [9], and dense optical flow field [4]. These methods are strongly based on silhouette extraction and therefore are not resistant to changing clothes or carrying luggage. It is also worth noting that most of them can only achieve correct results from a specific point of view, usually side view [4,5,7,8,9,16]. The recent research on model-free methods has focused on eliminating these drawbacks [6,10,11,12]. Model-based methods infer gait signature directly by modeling the underlying kinematics of human motion. The methods of this approach initially focused on using only static body parameters for recognition, such as stride length, which were updated over time [13,14]. Yam et al. [14] have extended this concept by analyzing the movement of the legs and the angles between them. In Ref. [15], the authors proposed a method that uses a motion-based model and elliptic Fourier descriptors to extract the key features of gait. Deng et al. [18] proposed a method that combines spatio-temporal and kinematic gait features. The fusion of two different features gives a comprehensive characterization of gait dynamics, which is less sensitive to walking conditions. In [21], the authors presented a gait recognition method that uses a 3D model of the human body and particle swarm optimization to obtain gait features. The number of obtained features was reduced using the multilinear principal component analysis (MPCA).

In the recognition process, various classification techniques are used; most often, they are classical methods such as k-nearest neighbors (kNN) [2,4,10,14,15,16,17,19,23], multilayer perceptron (MLP) [21], support vector machine [9,16,24] linear discriminant analysis (LDA) [16], and radial basis function neural networks [18]. The main focus in these papers is on developing descriptors that better describe gait features, rather than introducing new classification methods with better recognition ability. This is a traditional approach that is based on a clear separation between the descriptors and classifier model. On the other hand, in the papers of recent years, the introduction of new classification methods such as deep learning [11,20,22] or hybrid methods [12], in which the description and classification steps cannot be easily distinguished, is increasingly visible. In [20], the authors utilized a 3D convolutional neural network (CNN) and long short-term memory neural networks for training the classification models. They then used a grey wolf optimizer to tune the fusion parameters of each modality to boost the recognition performance of the system. Chao et al. [11] proposed a deep learning model called GaitSet. In this method, the CNN is used to extract frame-level features from each silhouette independently. Next, an operation called set pooling is used to aggregate frame-level features into a single set-level feature. In the end, a structure called horizontal pyramid mapping is used to map the set-level feature into a more discriminative space to obtain the final representation. The proposed method can extract spatial and temporal information more effectively than other methods regarding gait as a template or sequence. In turn, in the paper [12], the authors used a hybrid approach and combine the improved local coupled extreme learning machine and PSO for the classification process. A Gabor filter was used to extract gait features from the GEI and linear discriminant analysis was used to dimensionality reduction.

From the literature review, it is seen that most studies considering the traditional approach are devoted to developing gait descriptors, rather than introducing new classification methods. In this paper, we propose hybrid methods that combine the RDA and swarm intelligence techniques for gait recognition. In our approach, the PSO, GWO, and WOA are used to tune the observation weights and hyperparameters of the RDA model. To the best of our knowledge, the GWO and WOA algorithms have not been used before for this purpose. In the learning process, the confusion value is used as an objective function. The proposed methods are tested on a database of 414 gait cycles belonging to 32 different persons [21]. Summarizing, the main contributions of this paper can be stated as:proposing a combination of regularized discriminant analysis and particle swarm optimization for gait recognition,proposing a combination of regularized discriminant analysis and grey wolf optimization,proposing a combination of regularized discriminant analysis and whale optimization algorithm,comparing and improving the results obtained in the paper [21].

The structure of this article is as follows: Section 2 contains the description of the dataset and methods used in the recognition process. In particular, the structure of the gait recognition system, building classification models, and swarm intelligence methods are described. The experimental results are presented in Section 3. The results obtained by eight methods are presented, of which three were proposed by the authors. Section 4 contains the discussion of the achieved results. Finally, the conclusions are given in Section 5.

## 2. Material and Methods

### 2.1. Gait Dataset

The publicly available gait dataset (GPJATK) was used in the experiments [21]. The dataset consists of 166 data sequences (414 gait cycles) representing 32 people (10 women and 22 men). The sequences are divided into three subsets: 128 sequences (325 gait cycles) in which each of 32 individuals was dressed in his/her clothes; 24 sequences (58 gait cycles) in which 6 of 32 individuals (person #26–#31) changed clothes; and 14 sequences (31 gait cycles) in which 7 of the individuals (person #26–#32) had a backpack on his/her back. Each sequence contains video data (960x540@25fps) recorded using four calibrated and synchronized cameras and data from a markerless and marker-based motion capture systems. The synchronization between videos and motion capture data has been realized using Vicon MX Giganet. Our research is based on data obtained by a markerless motion capture system [25], which uses the annealed PSO algorithm in the motion capture process and data from four synchronized and calibrated cameras.

### 2.2. Gait Recognition System

A typical model-based system for gait recognition is presented in Figure 1. Such a system consists of gait capture, a feature extraction module, and a classifier. The objective of the system is to determine the identity of a gait sample using a database consisting of gait patterns from a set of known subjects. In the first step, one or more video-cameras are used to register the user’s image in a scene. In a preprocessing phase, image processing, i.e., background subtraction, body silhouette extraction, and edges extraction, is performed. The kinematic model of human motion is used in the next step to extract gait features that will define a gait signature. In the used gait dataset [21], each gait cycle is treated as a data sample represented by a third-order tensor with the dimension 32 × 11 × 3. The first dimension, equal to 32, is the average time of the gait cycle. The motion data was filtered using a moving average of length nine samples to the original data. The second dimension of the tensor is equal to the number of bones (excluding pelvis), i.e., 10 plus one element for storing a person height and distance between ankles. The third dimension relates to three angles, except the 11th vector that contains a person’s height, distance between ankles, and value of zero to maintain alignment with the rest of the vectors. Such a gait signature is then reduced using the MPCA algorithm [26]. The last element of the system is the classifier block that we focus in this article.

### 2.3. Building Classification Model for Gait Recognition

Building a gait classification model involves two main stages, which include training the model and testing it. For this purpose, the gait sequence database is divided into three sets: the training set, validation set, and test set (Figure 2). These three datasets are commonly used in different stages of the model building. Separating the dataset into these three subsets is used to avoid overfitting of the model. Initially, the model is fit on the training set, which is a set of observations used to fit the parameters of the RDA model. It should be noted that, in [21], only two sets were defined: training and testing. However, in the proposed approach, due to the optimization of the classifier parameters, an additional validation set is separated from the training data. The validation dataset is used for an evaluation of the fitted model while training the model’s hyperparameters. After building the model, the test set is used for testing, that is, for predicting the classifier’s output for data that has never been used in the training phase. Model training is carried out using one of the hybrid methods, in which swarm intelligence techniques optimize the observation weights and hyperparameters of the RDA. This problem is well suited for swarm optimization techniques because it creates a large search space to be explored. This search space is determined by the number of observation weights, the number of hyperparameters, and by the fact that all these variables are real values in the specified intervals. During the optimization, the following objective function, expressed as the confusion value, is used:(1)$$\mathrm{objective}\;\mathrm{function}\;\left(\mathrm{confusion}\right)=\frac{\mathrm{number}\;\mathrm{of}\;\mathrm{samples}\;\mathrm{misclassified}}{\mathrm{number}\;\mathrm{of}\;\mathrm{all}\;\mathrm{samples}}$$
where the samples are taken from the validation set. This objective function is minimized using one of the swarm intelligence methods, i.e., particle swarm optimization, grey wolf optimization, or whale optimization algorithm.

After building the RDA classifier, it is used to determine the correct classified ratio (CCR). The CCR is a ratio of correctly classified samples to the total number of samples in the test subset.

### 2.4. Regularized Discriminant Analysis

Linear discriminant analysis was developed by Sir Ronald Fisher in 1936 [27]. The original method proposed by Fisher was described for a 2-class problem, and it was in 1948 generalized as multi-class problems by Rao [28]. The LDA is a transformation technique used in statistics and machine learning to find linear combinations of features that separate classes of objects. The combinations obtained by this method may be used as:dimensionality reduction and feature extraction before classification,a linear classifier (considered in this paper).

The LDA consists of statistical properties of data calculated for each class. For a single variable, these are the mean and the variance of the variable. For multiple variables, these are the means and the covariance matrix. These statistical properties are estimated from the data and used to formulate an equation for making predictions. It should be emphasized that the use of the LDA is not associated with problems when the number of observations is greater than the dimension of each observation. Problems arise when the opposite is true, which makes the covariance matrix singular and cannot be inverted. To resolve this problem, instead of using the covariance matrix directly, a regularization of this matrix is used. This approach is applied to the regularized discriminant analysis method, in which the regularized covariance matrix ${\hat{\Sigma }}_{\gamma }$ is given by [29]:(2)$${\hat{\Sigma }}_{\gamma }=\left(1-\gamma \right)\hat{\Sigma }+\gamma I$$ where $\hat{\Sigma }$ is the covariance matrix, **I** is the identity matrix, and γ∈[0,1] is the amount of regularization. The RDA introduces regularization into the covariance matrix estimate, enabling a solution to be obtained and allowing different influences of variables on the classification model. In addition to the parameter *γ* the RDA model uses the parameter *δ* that acts as a threshold: if a model coefficient has the magnitude smaller than *δ* the RDA sets this coefficient to zero, and the corresponding predictor can be eliminated from the model.

The output of the RDA classifier $\hat{y}$ is calculated so as to minimize the classification cost [30]:(3)$$\hat{y}=\underset{y=1,\ldots ,K}{\arg\min}\sum_{k=1}^{K}\hat{P}\left(x|k\right)C\left(y|k\right)$$ where *K* is the number of classes, $\hat{P}\left(x|k\right)$ is the posterior probability of class *K* for observation *x*, C(y|k) is the cost of classifying an observation as *y* when its true class is *k*. The RDA used in this paper constructs weighted classifiers using the following scheme. Suppose **M** is an *N*-by-*K* class membership matrix such that $M_{nk}=1$ if observation *n* is from class *k*, $M_{nk}=0$, otherwise. The estimate of the class mean for weighted data with positive weights *w*_*n*_ is [30](4)$${\hat{\mu }}_{k}=\frac{\sum_{n=1}^{N}M_{nk}w_{n}x_{n}}{\sum_{n=1}^{N}M_{nk}w_{n}}$$


The estimate of the covariance matrix is(5)$$\hat{\Sigma }=\frac{\sum_{n=1}^{N}\sum_{k=1}^{K}M_{nk}w_{n}\left(x_{n}-{\hat{\mu }}_{k}\right){\left(x_{n}-{\hat{\mu }}_{k}\right)}^{T}}{1-\sum_{k=1}^{K}\frac{W_{k}^{\left(2\right)}}{W_{k}}}$$where $W_{k}=\sum_{n=1}^{N}M_{nk}w_{n}$ is the sum of the weights for class *k*, and $W_{k}^{\left(2\right)}=\sum_{n=1}^{N}M_{nk}w_{n}^{2}$ is the sum of squared weights per class.

### 2.5. Particle Swarm Optimization

A particle swarm optimization algorithm was developed by Kennedy and Eberhart [31]. This algorithm is based on the social behavior of organisms living in large groups. In the PSO, a group of agents called particles forms a swarm, where each particle represents a point in a multidimensional space. The particles explore this space in order to find the optimal solution. Each particle in the swarm is attracted both to its best position and the best position found by other particles. The best solution is obtained by minimizing the objective function.

Each particle has its position (**x**) and velocity (**v**). The velocity **v**_*k*_ of the *k*th particle is determined using the following equation [31]:(6)$$v_{k}\left(t+1\right)=\omega v_{k}\left(t\right)+c_{1}r_{1}\left({\mathrm{pbest}}_{k}\left(t\right)-x_{k}\left(t\right)\right)+c_{2}r_{2}\left(\mathrm{gbest}\left(t\right)-x_{k}\left(t\right)\right)$$ where *t* is the current iteration number, *ω* is the inertia weight, **r**_1_, **r**_2_ are vectors of random numbers in the range [0,1], *c*_1_ is the cognitive coefficient, and *c*_2_ is the social coefficient. It is seen that the update of the velocity is a weighted sum of the previous velocity $v_{k}\left(t\right)$, the difference between the current position and the personal best position (**pbest**), and the difference between the current position and the global best position (**gbest**). The position **x**_*k*_ of the *k*th particle is updated according to the equation(7)$$x_{k}\left(t+1\right)=x_{k}\left(t\right)+v_{k}\left(t\right)$$

After updating the velocity and the position, the objective function is calculated to determine the personal and global positions.

### 2.6. Grey Wolf Optimization

A grey wolf optimizer is another swarm intelligence technique used to solve optimization problems [32]. The GWO algorithm is inspired by the behavior and hierarchy of grey wolves in nature, searching for the optimal way to attack their prey. In the hierarchy of grey wolves, the most dominating is alpha (α), which leads the entire group. The other wolves are beta (β) and delta (δ), which help to control the rest of the wolves considered as omega (ω). The omega wolves have the lowest ranking in the hierarchy. The main phases of grey wolf hunting are: (a) tracking, chasing, and approaching; (b) chasing, encircling, and harassing; (c) attacking.

#### 2.6.1. Encircling Prey

Grey wolves encircle the prey during the hunt, which can be mathematically modeled by the following equation [32]:(8)$$X\left(t+1\right)=X_{p}\left(t\right)-A\cdot D$$
where
(9)$$D=|C\cdot X_{p}\left(t\right)-X\left(t\right)|$$
and X(t) is the current position of a grey wolf at iteration *t*, $X_{p}\left(t\right)$ is the position of the prey, and “·” is an element-by-element multiplication. The coefficient vectors A, C are determined as follows: (10)$$\begin{array}{cc} A & =2a\cdot r_{1}-a \end{array}$$(11)$$\begin{array}{cc} C & =2r_{2} \end{array}$$
where components of a are linearly decreased from 2 to 0 through iterations and $r_{1}$, $r_{2}$ are random vectors in [0,1].

#### 2.6.2. Hunting

In the GWO, we assume that the α, β, and δ are the best solutions for the entire population. Therefore, the other wolves should update their position according to the positions of the three agents. The following formula is used to calculate the positions of search agents [32]:(12)$$X\left(t+1\right)=\frac{1}{3}\left(X_{1}\left(t\right)+X_{2}\left(t\right)+X_{3}\left(t\right)\right)$$
where
(13)$$\begin{array}{cc} X_{1} & =X_{\alpha }\left(t\right)-A_{1}\cdot D_{\alpha } \end{array}$$
(14)$$\begin{array}{cc} X_{2} & =X_{\beta }\left(t\right)-A_{2}\cdot D_{\beta } \end{array}$$
(15)$$\begin{array}{cc} X_{3} & =X_{\delta }\left(t\right)-A_{3}\cdot D_{\delta } \end{array}$$
and
(16)$$\begin{array}{cc} D_{\alpha } & =|C_{1}\cdot X_{\alpha }-X| \end{array}$$
(17)$$\begin{array}{cc} D_{\beta } & =|C_{2}\cdot X_{\beta }-X| \end{array}$$
(18)$$\begin{array}{cc} D_{\delta } & =|C_{3}\cdot X_{\delta }-X| \end{array}$$

The vectors $A_{1}$, $A_{2}$, $A_{3}$ are obtained using Equation (10), while $C_{1}$, $C_{2}$, $C_{3}$ are obtained using Equation (11).

#### 2.6.3. Attacking Prey (Exploitation) and Search for Prey (Exploration)

The grey wolves start the attack, once the prey stops moving. To model the process of approaching the prey, the GWO linearly decrease all the values of a from 2 to 0 according to the equation [32]
(19)$$a=2-\frac{2t}{T_{max}}$$
where $T_{max}$ is the total number of iterations of the algorithm. The change of a affects the coefficient vector A, which controls the behavior of search agents. If |A|<1, the wolf will move towards the prey; on the other hand, if |A|>1, the wolf will diverge from the prey in order to find new better prey. In addition, the vector C which contains a random value in the range [0,2] is employed to help the algorithm to avoid being trapped in the local optima.

### 2.7. Whale Optimization Algorithm

The whale optimization algorithm is a nature-inspired metaheuristic technique for solving optimization problems [33]. This algorithm mimics the social behavior of humpback whales realized in the bubble-net hunting strategy. The WOA is based on three operators to simulate the search for prey, encircling prey, and bubble-net foraging.

#### 2.7.1. Encircling Prey

Humpback whales encircle pray after recognizing its position. In this phase, the search agents attempt to change their locations towards the best search agents. This behavior is expressed by the following formula [33]:(20)$$X\left(t+1\right)=X^{*}\left(t\right)-A\cdot D$$
where
(21)$$D=|C\cdot X^{*}\left(t\right)-X\left(t\right)|$$
and X(t) is the current position vector at iteration *t*, $X^{*}\left(t\right)$ is the best position obtained so far, “·” is an element-by-element multiplication. The coefficient A, C are determined from the formulas:(22)$$\begin{array}{cc} A & =2a\cdot r_{1}-a \end{array}$$(23)$$\begin{array}{cc} C & =2r_{2} \end{array}$$
where components of a are linearly decreased from 2 to 0 through iterations and $r_{1}$, $r_{2}$ are random vectors in [0,1].

#### 2.7.2. Bubble-Net Attacking (Exploitation Phase)

In this phase, humpback whales swim around the prey within a helix-shaped path. To model this behavior, it is assumed that there is a 50% chance to choose between the shrinking encircling or spiral movements [33]:(24)$$X\left(t+1\right)=\left\{\begin{array}{cc} X^{*}\left(t\right)-A\cdot D\; & \mathrm{if}\;p<0.5\\ D^{*}\cdot \exp\left(bk\right)\cdot \cos\left(2\pi k\right)+X^{*}\left(t\right) & \mathrm{if}\;p\geq 0.5 \end{array}\right.$$
where $D^{*}=\left|X^{*}\left(t\right)-X\left(t\right)\right|$ is the distance of the *i*th whale to the prey, *b* defines the shape of the spiral, *k* is a random number in [−1,1], and *p* is a random number in [0,1].

#### 2.7.3. Search for Prey (Exploration Phase)

In this phase, humpback whales search the pray according to the position of each other. The location of a search agent is calculated according to randomly selected search agent rather than the best search agent as in the exploitation phase. The mathematical model is determined as follows [33]:(25)$$X\left(t+1\right)=X_{rand}\left(t\right)-A\cdot D$$
where
(26)$$D=|C\cdot X_{rand}\left(t\right)-X\left(t\right)|$$
and $X_{rand}$ is the random position vector chosen from the current population.

### 2.8. Integration of Swarm Intelligence Techniques with Regularized Discriminant Analysis

The idea of integrating swarm intelligence methods with the RDA classifier is shown in Figure 3. This figure presents in the form of a block diagram main stages of optimization of the RDA model using swarm algorithms and the method of determining the objective function. The task of particle swarm optimization, grey wolf optimization, or whale optimization algorithm is to select the RDA parameters, which are [30]:$w_{1},w_{2},\ldots ,w_{n}$ — the observation weights,δ — the linear coefficient threshold,γ — the parameter for regularizing the covariance matrix of the predictors,where *n* is the number of observations in the training set. For this purpose, the observation weights and two hyperparameters δ, γ are placed as elements of the agent (candidate) vector, which has the form
(27)$$\overset{-}{\underset{-}{|\;w_{1}\left|\;w_{2}\left|\;\ldots \;\right|\;w_{n}\left|\;\delta \;\right|\gamma \;\right|}}$$

At the beginning of the algorithm, the agents are initialized. In the next step, the value of the objective function for all agents is calculated. Then, the stop condition is checked, if it is not reached, the agents are updated and the value of the objective function is recalculated. The swarm optimization algorithm generates many hypothetical solutions (which are represented by agents) and the best solution is selected in the optimization process. These operations are repeated until the stop condition is reached. In the objective function block, the RDA model is determined for the parameters proposed by the agent and for the training data. The output of this model is then calculated on the validation data and the value of the objective function is determined based on formula (Equation 1). The weights $w_{1},w_{2},\ldots ,w_{n}$, hyperparameters δ and γ of the RDA classifier are limited during optimization in given ranges (see Section 3). The algorithm returns the optimal parameters of the RDA classifier in the best agent vector. This result is included in the block ’Return the global best solution’ in Figure 3. On this basis, the objective function value for the test data are calculated.

All the proposed hybrid methods have been implemented in Matlab equipped with additional toolboxes. Using the function fitcdiscr from the Statistics and Machine Learning Toolbox [30], the RDA classifier model is created, while the function predict from the same package is used to determine class predictions (Figure 3). The PSO method has been implemented using the function particleswarm from the Global Optimization Toolbox [34] and the GWO and WOA methods using software developed by Mirjalili [32,33].

## 3. Results

The division of the gait dataset into training, validation, and test sets in four experiments is presented in Table 1. In the first experiment (Set #1) and the fourth experiment (Set #4), all persons in the collection were wearing clothes number 1. In the second experiment (Set #2), persons in the training and validation sets were in clothing number 1, while in the test set in clothing number 2. In the third experiment (Set #3), persons in the training and validation sets were in clothing number 1, while in the test set they wore a backpack. The number of identities in training/validation/test subsets was: in Set #1—32/32/32, Set #2—32/32/6, Set #3—32/32/7, and Set #4—32/32/32. The samples were not repeated in the subsets. The procedure of separating the samples from the training set to the validation set was as follows: for Sets #1 and #4, one sample was taken for each class (the last sample was always taken), for Sets #2 and #3, two samples were taken for each class because there were more training data (the last two samples for each class were taken). Experiments 1–3 were performed in such a way that the classification model and its testing error were determined 10 times, and then the average of the results was calculated. In the fourth experiment (Set #4), the 10-fold cross-validation was performed to obtain an average score. In this method, the original dataset is partitioned into 10 equal size subsets. A single subset is retained as the test data, and the remaining nine subsets are used as training data. The cross-validation process is repeated 10 times (the folds) for the test data, and the 10 results are averaged. In the RDA-PSO, RDA-GWO, and RDA-WOA methods, the number of agents was equal to 30, and the number of iterations was equal to 25. These parameters of optimization techniques were selected experimentally. The weights of the observations were limited to the range $\left[10^{-8},1\right]$, while the hyperparameters δ and γ were limited to the range [0,1].

Table 2 contains the correct classified ratio of the gait recognition for the considered methods. These are four classical methods (kNN [35], NB [35], support vector machines with sequential minimal optimization (SMO) [36], and MLP [35]) taken from [21], linear discriminant analysis (non-regularized) [27], and three proposed hybrid methods. Figure 4 presents the confusion matrices for the best models in the considered experiments. These matrices show the percentage of correct class recognition by the arrangement of the colored elements. The closer the color is to dark red on the diagonal, the more accurate the class recognition. The colors changing from dark red to white mean worse and worse recognition.

## 4. Discussion

To compare the results, the classical methods taken from the paper [21] and the LDA were considered. This comparison with the methods proposed by the authors (RDA-PSO, RDA-GWO, RDA-WOA) is provided in Table 2. It can be seen that, in Experiment 1 and Experiment 2, the RDA-PSO method proposed by the authors has the highest CCR index. In Experiment 3, which considered clothing with a backpack, three of the analyzed methods (LDA, RDA-GWO, RDA-WOA) achieved the same CCR at the level of 94%. In the final fourth experiment, the method proposed by the authors was again the best, in this case combining RDA with GWO. It should also be noted that the LDA obtained the worst result of all methods for Experiment 1; this is most likely due to insufficient training data. However, the proposed methods obtained very good results for this experiment, at the level of about 86–87%. The use of the regularized discriminant analysis combined with swarm intelligence techniques improved the quality of gait recognition. The developed methods outperformed the classical methods and improved the recognition results achieved by the best of them (MLP) by 6 to 16%, depending on the experiment. When comparing the proposed hybrid methods between each other, it should be noted that the results are inconclusive and it seems that these methods are equivalent in this application.

When analyzing the confusion matrix shown in Figure 4, it can be seen that most people are recognized with high accuracy, but there are classes with which the methods have difficulties. For example, in Experiment 1 (RDA-PSO method) for classes 6, 15, 20, 21, and 25, the recognition efficiency drops to 40–60%. It is most likely caused by too little training data (about five gait cycles for each class). When 10-fold cross-validation is used (about nine gait cycles for each class), the recognition efficiency increases significantly. It should also be noted that the data used in experiments were recorded with a markerless motion capture system, which is not perfect and generates noise [21]. It certainly has an impact on the achieved results. For Experiment 2, for which learning and testing were performed on the sequences for which the clothes were changed, a significant deterioration in the results for persons #30 and #31 can be observed. When analyzing these video sequences, it can be noticed that the heavy shoes changed to sandals. This could be the cause of the deterioration in the results in this experiment. On the other hand, for person #29, there was a change of footwear from sports shoes to shoes with a heel, and still 100% detection rate was achieved.

## 5. Conclusions

The hybrid methods that combine regularized discriminant analysis and swarm intelligence techniques for gait recognition have been proposed. In the presented approach, particle swarm optimization, grey wolf optimization, and whale optimization algorithm are used. These techniques optimize the observation weights and hyperparameters of the regularized discriminant analysis. The proposed methods were compared with five methods found in the literature. In the learning process, the confusion value was used as an objective function. The conducted experiments on the GPJATK dataset proved the validity of the proposed concept. Future work will focus on improving the proposed concept by replacing the MPCA method with another method of dimensionality reduction. Moreover, some work will be carried out to add new features to gait signatures. 

## Figures and Tables

**Figure 1 sensors-20-06794-f001:**
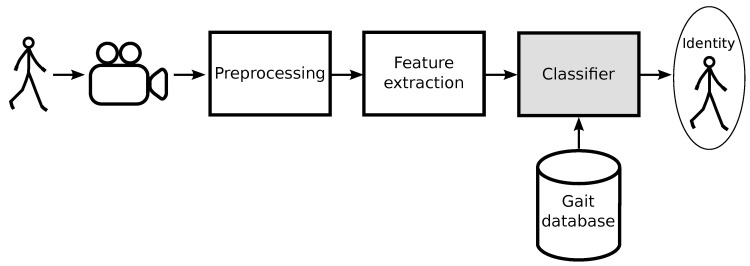
Structure of a gait recognition system.

**Figure 2 sensors-20-06794-f002:**
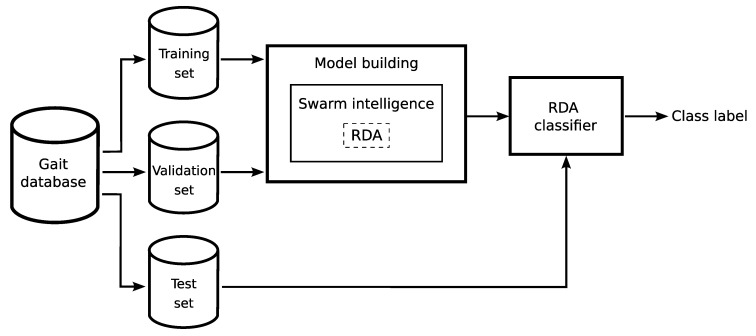
The idea of building the RDA classification model.

**Figure 3 sensors-20-06794-f003:**
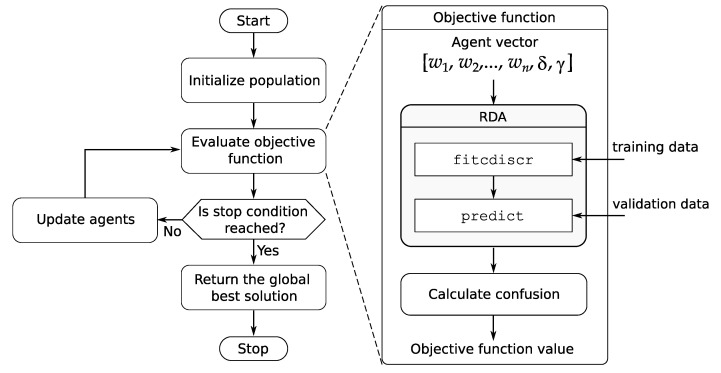
The idea of integration of swarm intelligence techniques with the RDA.

**Figure 4 sensors-20-06794-f004:**
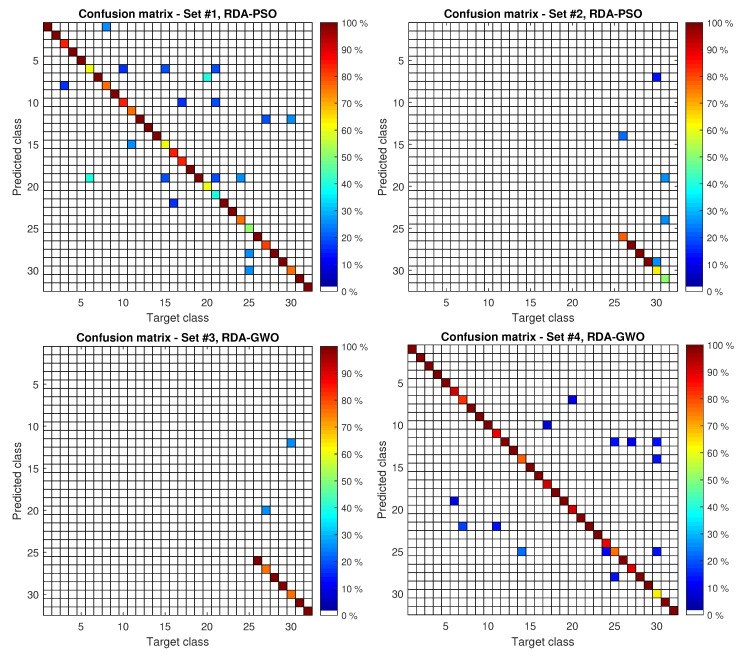
Confusion matrices for best models in each experiment.

**Table 1 sensors-20-06794-t001:** Division of the gait dataset into training, validation, and test sets.

Experiment	Subset	Classical Methods	Hybrid Methods
1: Set #1	train	169	137
	validation	–	32
	test	156	156
2: Set #2	train	325	261
	validation	–	64
	test	58	58
3: Set #3	train	325	261
	validation	–	64
	test	31	31
4: Set #4	train	90% (≈293)	80% (≈261)
	validation	–	10% (≈32)
	test	10% (≈32)	10% (≈32)

**Table 2 sensors-20-06794-t002:** Correct classified ratio [%].

Experiment	kNN [21]	NB [21]	SMO [21]	MLP [21]	LDA	RDA-PSO	RDA-GWO	RDA-WOA
1: Set #1	47.44	55.77	67.95	80.13	45.51	87.05	86.28	86.92
2: Set #2	37.93	56.90	63.79	75.86	79.31	85.34	84.48	84.48
3: Set #3	38.71	70.97	67.74	77.42	93.55	88.39	93.55	93.55
4: Set #4	56.92	79.69	84.31	89.85	91.99	95.07	95.39	95.09

The best result is marked in bold font.

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
