# Peer review of "Combined Regularized Discriminant Analysis and Swarm Intelligence Techniques for Gait Recognition"

_sensors, 2020, doi:10.3390/s20236794_

Round 1

Reviewer 1 Report

The authors main contribution is the use of regularized discriminant analysis (RDA), minimizing the objective function using one of the swarm intelligence methods. But there is no specific reference to RDA, nor the explanation over the objective function being reduce for the swarm intelligence methods.

In the figure 3, the results are presented on the confusion matrices, but the set 3 and the set 2, don’t need show all the classes, because not all classes are used.

Author Response

We want to thank you for your comments that helped us to improve the manuscript. We hope that you will now find it suitable for publication in Sensors. The revisions in the manuscript are highlighted using the red font for your convenience of re-reviewing.  

Point 1: The authors main contribution is the use of regularized discriminant analysis (RDA), minimizing the objective function using one of the swarm intelligence methods. But there is no specific reference to RDA, nor the explanation over the objective function being reduce for the swarm intelligence methods. 

Response 1: The objective function is given in (1). 
A more detailed description of the RDA is given in section 2.4, and the combination of RDA and swarm intelligence techniques is given in new section 2.8 (“Integration of swarm intelligence techniques with regularized discriminant analysis”). This section also describes the implementation of the proposed methods in Matlab. 

Point 2: In the figure 3, the results are presented on the confusion matrices, but the set 3 and the set 2, don’t need show all the classes, because not all classes are used. 

Response 2: All classes must be shown because the algorithm can recognize a given sample as any of 32 classes, e.g., in confusion matrix for Set #2, class 30 was misclassified as 7. Moreover, we wanted the confusion matrices for all experiments to look similar. 

Reviewer 2 Report

In their Manuscript, “Combined regularized discriminant analysis and swarm intelligence techniques for gait recognition”, the Author compare the performance of a new method for gait recognition, based on the use of both RDA and one of three different swarm intelligence techniques (namely, the PSO, the GWO, and the WOA) with those of five classical methods (4 of them selected based on a previously published paper). According to their results, the combination of RDA and one SIT lead to better results. The manuscript is interesting and worth publication. However, for the sake of readability.  the reporting should be increased. The main points are reported below.

Introduction

  1. (bulleted list,  lines 70-75). Please review the bulleted list: in fact, the first line is general, the second and the third refer to two over three analyzed technique (why was the PSO omitted?) and, therefore, they should be in a lower level; in the last line, “improving” should be replaced by “comparing”. In general, however, I do not believe that “the use of…”, or “testing” are really a contribution. I suggest rewriting the entire list.  
  2. (lines 76-81). Section or sect.? Please be consistent.

Material and methods

  1. (line 92). Does the “markerless motion capture system” refer to the four cameras? Please specify in the text.
  2. How were the cameras positioned with respect to the subjects? Data from different cameras were considered as independent observations? In this case, did the orientation of the cameras impact the results?
  3. Otherwise, how the use of a single camera could impact the results, according to the Authors’ opinion?
  4. (line 109): please better describe the use of three sets (training, validation, and tests).
  5. Please better describe the integration of the swarm intelligence methods in the RDA
  6. (Line 130): …for example, we can use regularized discriminant analysis. This sentence is not clear. Generally speaking, the Linear Discriminant Analysis (used as a classical model) and the Regularized discriminant analysis (used in addition to the swarm intelligence techniques) should be better discriminated and described.
  7. Why the authors did not describe, at least shortly, the “five classical models”?
  8. Swarm intelligence techniques: please describe all the phases (e.g.: in the GWO, the chasing is missing; harassing and attacking are parts of the hunting?; moreover, in the WOA, the order of the phases is different).

Results.

  1. (Line 177) Please describe the 10-fold CV. Moreover, in Table 1, the 10-fold CV can be considered as an experiment, as the set#1, set#2, and set#3?
  2. It is unclear why the CCR in the set#2 and set#3 is higher when compared to set#1. Owing to the nature of the experiment, shouldn't it be the other way around? Is this result related to the sample size? Could the Authors please comment on that?
  3. Relating to the latter question, in the Authors’ opinion, aren’t 5 and 6 target classes too few for the analysis?

Discussion

  1. In my opinion, the discussion section should be improved. Specifically, it is not clear what combined method, among the three, is the best, and why.  

Author Response

We want to thank you for your comments that helped us to improve the manuscript. We hope that you will now find it suitable for publication in Sensors. The revisions in the manuscript are highlighted using the red font for your convenience of re-reviewing. 

Introduction 

Point 1: (bulleted list,  lines 70-75). Please review the bulleted list: in fact, the first line is general, the second and the third refer to two over three analyzed technique (why was the PSO omitted?) and, therefore, they should be in a lower level; in the last line, “improving” should be replaced by “comparing”. In general, however, I do not believe that “the use of…”, or “testing” are really a contribution. I suggest rewriting the entire list. 

Response 1: We have made the following changes to the bulleted list: 

  • we changed the first bullet so that it now only applies to the PSO algorithm: “proposing a combination of regularized discriminant analysis and particle swarm optimization for gait recognition,”. We deliberately omitted the PSO in the first draft of the article and closed it in the general term along with other swarm algorithms. We did it because, as far as we know, the RDA+SIT combination was not used for gait recognition, while the RDA+GWO and RDA+WOA algorithms are our proposals. In the current version of the article, it is noted that RDA+PSO is our concept regarding gait recognition, while RDA+GWO and RDA+WOA are completely new algorithms. 
  • we removed the fourth bullet; 
  • in the last bullet, we added the phrase "comparing" and now it has the following form: “comparing and improving the results obtained in the paper”. We want to leave the word "improving" because we believe that it is also our contribution and the improvement of the results, we obtained is worth noting. 

Point 2: (lines 76-81). Section or sect.? Please be consistent. 

Response 2: Corrected to "Section". 

Material and methods 

Point 3(line 92). Does the “markerless motion capture system” refer to the four cameras? Please specify in the text. 

Response 3 Obtaining gait data is not part of our work, but we use data from the GPJATK database [21].  
The structure of the 3D data is described in section 2.2. The 3D data describing human movement was obtained using a markerless motion capture system. This system estimated 3D data by processing video sequences from four synchronized and calibrated cameras. 
We changed the mentioned sentence to: “Our research is based on data obtained by a markerless motion capture system [25], which uses the annealed PSO algorithm in the motion capture process and data from four synchronized and calibrated cameras.” 

Point 4: How were the cameras positioned with respect to the subjects? Data from different cameras were considered as independent observations? In this case, did the orientation of the cameras impact the results? 

Response 4: 1) The layout of the cameras is presented in the paper describing the GPJATK dataset [21] in Fig. 1. Every person walked from right to left, then from left to right, and afterward on the diagonal from upper-right to bottom-left and from bottom-left to upper-right corner of the scene. The entire walk was recorded by all cameras.   
2) No, data from all cameras was considered simultaneously by the markerless motion capture system. 

Point 5: Otherwise, how the use of a single camera could impact the results, according to the Authors’ opinion? 

Response 5: Recognition performance depends on the quality of 3D human motion data, and obtaining accurate 3D data from sequences from a single camera is very difficult to achieve. Thus, the use of a motion capture system based on data from only one camera will have a significant impact on the achieved recognition results. 

Point 6: (line 109): please better describe the use of three sets (training, validation, and tests). 

Response 6: These sets are described in more detail in section 2.3. 

Point 7: Please better describe the integration of the swarm intelligence methods in the RDA. 

Response 7: The integration of the swarm intelligence methods with the RDA is described in more detail in new section 2.8 (“Integration of swarm intelligence techniques with regularized discriminant analysis”). 

Point 8: (Line 130): …for example, we can use regularized discriminant analysis. This sentence is not clear. Generally speaking, the Linear Discriminant Analysis (used as a classical model) and the Regularized discriminant analysis (used in addition to the swarm intelligence techniques) should be better discriminated and described. 

Response 8: The description has been corrected. The RDA method has been described in more detail. 

Point 9: Why the authors did not describe, at least shortly, the “five classical models”? 

Response 9: The results achieved by these methods were taken from [21] and used only to compare the results. Additionally, we have added citations to the papers describing these methods. 

Point 10: Swarm intelligence techniques: please describe all the phases (e.g.: in the GWO, the chasing is missing; harassing and attacking are parts of the hunting?; moreover, in the WOA, the order of the phases is different). 

Response 10: Section 2.6 describes the general concept of the GWO algorithm based on grey wolf hunting. The main phases of grey wolf hunting are a) tracking, chasing, and approaching; b) chasing, encircling, and harassing; c) attacking. On their basis, the article [31] proposes a mathematical model of the GWO algorithm including encircling prey, hunting, attacking prey (exploitation), and search for prey (exploration). In fact, we omitted two phases of the algorithm (attacking prey and search for prey). The description of these phases has been added to the revised version of the article. 
The order of description of the individual phases of GWO and WOA was taken from the articles in which the above-mentioned algorithms were proposed. 

Results 

Point 11(Line 177) Please describe the 10-fold CV. Moreover, in Table 1, the 10-fold CV can be considered as an experiment, as the set#1, set#2, and set#3? 

Response 11: The 10-fold CV is described in section 3. As suggested, the name of the 4th experiment has been changed to “Set #4”. 

Point 12: It is unclear why the CCR in the set#2 and set#3 is higher when compared to set#1. Owing to the nature of the experiment, shouldn't it be the other way around? Is this result related to the sample size? Could the Authors please comment on that? 

Response 12: For the proposed hybrid methods, the worst results were obtained for Set #2. The relationship mentioned by the reviewer occurred only for Set #3 and Set #1.  
In our opinion, comparing the results of individual experiments seems questionable because each experiment has a different amount of data. Our task was to develop new classification methods that give better results than those known in the literature for a given database. 

Point 13: Relating to the latter question, in the Authors’ opinion, aren’t 5 and 6 target classes too few for the analysis? 

Response 13: In all experiments, there are 32 classes in the training set, while in the test sets there are 6 classes (Set #2) and 7 classes (Set #3), for Set #1 and #4 there are 32 classes. We agree that the number of target classes may be too small for a complete analysis, but they also give some information on the effectiveness of the method. 

Discussion 

Point 14: In my opinion, the discussion section should be improved. Specifically, it is not clear what combined method, among the three, is the best, and why. 

Response 14: The following sentence was added in Discussion: “When comparing the proposed hybrid methods, it should be noted that the results are inconclusive and it seems that these methods are equivalent in this application.” 

Reviewer 3 Report

Objectives and goals are clearly pointed out. The study is properly designed regarding the objectives of the study

Author Response

We want to thank you for your review.  

Reviewer 4 Report

Summary:

In this paper, authors propose a system for 3D markerless gait recognition consisting of regularized discriminant analysis (RDA) and three swarm intelligence techniques: particle swarm optimization (PSO), grey wolf optimization (GWO) and whale optimization algorithm (WOA). The system is evaluated on their GPJATK dataset, of 32 people and 166 data sequences, which is available for download from their laboratory web page. One of the objectives of this paper is to improve the results of their 2019 MTAP paper where they introduce the GPJATK dataset along with four baseline models kNN, NB, SMO and MLP. Results indicate a roughly 93-95% correct classification rate on the full dataset, 85-87% on sub-dataset of specific clothing setups.

Detailed comments:

Line 13: This sentence has been taken from Jain, Flynn, Ross: Handbook of Biometrics, 2008.
Line 62: “most studies are devoted to developing gait descriptors, rather than introducing new classification methods”. In principle, descriptors and classifiers are nowadays interpreted as the same thing. Training a classifier is equivalent to learning descriptors.
Line 106: “The third dimension relates to three angles, except the 11th vector that contains a person’s height and distance between ankles.“ — 1st value is height, 2nd value is ankle distance, and what is the 3rd value?
Figure 2: RDA is not part of the swarm intelligence model but rather a dimensionality reduction applied before a particular optimization technique. Also, classifier outputs a class label and not the correct classification ratio.
Sections 2.4: It would be better if you included a detailed description of RDA. Please also give a comment on why you use RDA instead of PCA or LDA.
Sections 2.5—2.7: IMPORTANT. It is not clear how solving these optimization problems corresponds to training the classifiers. Please create and describe in detail a reduction function from the output of the swarm techniques onto classifier parameters.
Table 2: Please comment on why you didnt evaluate also SVM or any CNN-based model mentioned in related work.

Decision:

Major revision. I will need to see this paper improved before recommending to accept.

Author Response

We want to thank you for your comments that helped us to improve the manuscript. We hope that you will now find it suitable for publication in Sensors. The revisions in the manuscript are highlighted using the red font for your convenience of re-reviewing. 

Point 1: Line 13: This sentence has been taken from Jain, Flynn, Ross: Handbook of Biometrics, 2008. 

Response 1: The citation of this handbook was added in line 14. 

Point 2Line 62: “most studies are devoted to developing gait descriptors, rather than introducing new classification methods”. In principle, descriptors and classifiers are nowadays interpreted as the same thing. Training a classifier is equivalent to learning descriptors. 

Response 2Thank you for the pertinent observation. We agree that in the recently popular end-to-end deep learning models, the classifier and the way the observed scene is described are merged. Consequently, while the network is trained, the description and classification steps cannot be easily distinguished.  However, since we consider a traditional approach in which the descriptor is separated from the classifier, the descriptor describes the gait (see Section 2.2), and the obtained features are classified. In our work, we focus on improving the performance of classification methods that are suitable for a specific task. 

We introduced the following changes in the paper: 
1) Lines 46-51: “The main focus in these papers is on developing descriptors that better describe gait features, rather than introducing new classification methods with better recognition ability. This is a traditional approach that is based on a clear separation between the descriptors and classifier model. On the other hand, in the papers of recent years, the introduction of new classification methods such as deep learning [11,20,22] or hybrid methods [12], in which the description and classification steps cannot be easily distinguished, is increasingly visible. 
2) Line 64: “From the literature review, it is seen that most studies considering the traditional approach are devoted to developing gait descriptors, rather than introducing new classification methods.” 

Point 3Line 106: “The third dimension relates to three angles, except the 11th vector that contains a person’s height and distance between ankles.“ — 1st value is height, 2nd value is ankle distance, and what is the 3rd value? 

Response 3: The third value is 0 to maintain alignment with the rest of the vectors. This sentence has been changed as follows: 
“The third dimension relates to three angles, except the 11th vector that contains the person’s height, distance between ankles, and value of zero to maintain alignment with the rest of the vectors. “ 

Point 4Figure 2: RDA is not part of the swarm intelligence model but rather a dimensionality reduction applied before a particular optimization technique. Also, classifier outputs a class label and not the correct classification ratio. 

Response 4 You are right that the RDA is not part of swarm intelligence algorithms, but on the other hand, the RDA is called in the objective function of these algorithms, which is explained in detail in new section 2.8 and figure 3. For this reason, the RDA block was placed in the swarm intelligence block in figure 2. In our article, the RDA is not used for dimension reduction but as a classifier (see Matlab function ‘fitcdiscr’) described at the end of section 2.8. 
Classifier output in figure 2 has been changed to "class label". 

Point 5Sections 2.4: It would be better if you included a detailed description of RDA. Please also give a comment on why you use RDA instead of PCA or LDA. 

Response 5A more detailed description of the RDA is given in section 2.4, and the combination of RDA and swarm intelligence techniques is given in new section 2.8 (“Integration of swarm intelligence techniques with regularized discriminant analysis”). This section also describes the implementation of the proposed methods in Matlab. 
In addition to the RDA, we also used the LDA, but this method performed worse than the RDA (except for Set #3, where it had the same result) (see table 2). The gait data [21] were reduced by the MPCA method, which is an extension of the PCA. It has been mentioned at the end of section 2.2. 

Point 6Sections 2.5—2.7: IMPORTANT. It is not clear how solving these optimization problems corresponds to training the classifiers. Please create and describe in detail a reduction function from the output of the swarm techniques onto classifier parameters. 

Response 6The objective function is described in detail in section 2.3 (equation (1)). The output of swarm techniques is the vector containing observation weights and hyperparameters delta and gamma of the RDA classifier. The method of integration of swarm techniques with the RDA is described in new section 2.8 and presented in the form of a block diagram in figure 3. 

Point 7Table 2: Please comment on why you didnt evaluate also SVM or any CNN-based model mentioned in related work. 

Response 7In fact, Table 2 contains results of the modified SVM algorithm that uses the sequential minimal optimization for training and is implemented in the WEKA library. The name SMO is used to follow the naming convention from the WEKA library and article with the gait database [21]. To avoid any confusion we have added the following explanation in the Results section: “... support vector machines with sequential minimal optimization (SMO) [36], …". 
As for CNN, the paper [21] contains the results achieved by this kind of network for Set #2 and Set #3, but due to the poor results (about 65%) and the lack of results for Set #1 and Set #4, we decided not to use them at our article. 

Round 2

Reviewer 4 Report

I thank the authors for their quick response.
4/7 of my points have been addressed sufficiently.
But two important points need to be completed.
Thank you for bearing with me.

Between lines 122--123: “The error calculated on the validation set can be used to avoid overfitting to the training set.” — High validation score doesnt imply overfitting. What does prevent overfitting is separating the dataset into train/valid/test sets alongside the identity labels, that is, all gait sequences of each person fall exactly into one of these 3 sets. In other words, an instance of overfitting is when you use one sequence of a person for training and another sequence of the same person for testing. Can you please write in the paper how many identities appear in each of these 3 sets and if there is an overlap? Also, it would be very useful to add this data separation scheme in your dataset download page.

Line 188: Technical implementation details should rather be documented in a GitHub readme file.

Section 2.4: IMPORTANT. First, Figure 3 shows that input to RDA is the action vector [w_1,...,w_n,delta,gamma]. It is clear how gamma is used. However, it is not clear how you use the remaining w_1,...,w_n and delta. Second, how does the classifier compute the class label? Here I need a function that takes as input the trained classifier parameters (I guess sigma?) and the gait sequence, and outputs the predicted class label.

Equation 3: The variables pbest and gbest are defined but never used in the rest of the paper. As they represent an iteratively improving solution, it is intuitive to me that they play a role in deriving the classifier parameters. Can you please put them in the corresponding equations?

Response 4: I see it now. Then maybe I suggest two slight changes in the Figure 2: [1] Place RDA away from the swarm intelligence box, but still inside the model building box, and add an arrow from it to the swarm intelligence box; [2] Include training set and validation set inside the model building box? But if I am wrong then please disregard this.

Response 6: IMPORTANT. This is not what I meant. Between lines 122--123 you write that “Model training is carried out using one of the hybrid method, in which swarm intelligence techniques optimize the observation weights and hyperparameters of the RDA.” -- and it is substantial that you give the exact function to calculate the agent vector from intermediate positions (outputs) of the swarm techniques.

Author Response

We want to thank you for your comments that helped us to improve the manuscript. We hope that you will now find it suitable for publication in Sensors. The revisions in the manuscript are highlighted using the blue font for your convenience of re-reviewing.

Point 1: Between lines 122--123: “The error calculated on the validation set can be used to avoid overfitting to the training set.” — High validation score doesnt imply overfitting. What does prevent overfitting is separating the dataset into train/valid/test sets alongside the identity labels, that is, all gait sequences of each person fall exactly into one of these 3 sets. In other words, an instance of overfitting is when you use one sequence of a person for training and another sequence of the same person for testing. Can you please write in the paper how many identities appear in each of these 3 sets and if there is an overlap? Also, it would be very useful to add this data separation scheme in your dataset download page. 

Response 1: This sentence was changed and placed earlier in section 2.3: “Separating the dataset into these three subsets is used to avoid overfitting of the model. “ 

The following explanation was added in section 3: 

“The number of identities in training/validation/test sets was: in Set #1 -- 32/32/32, in Set #2 -- 32/32/6, Set #3 -- 32/32/7, and Set #4 -- 32/32/32. The samples were not repeated in the subsets. The procedure of separating the samples from the training set to the validation set was as follows: for Sets #1 and #4, one sample was taken for each class (the last sample was always taken), for Set #2 and #3, two samples were taken for each class because there were more training data (the last two samples for each class were taken).”  

Modifying the GPJATK dataset is problematic, but we believe that the sample selection process has been described in such detail that anyone can repeat the experiments. 

Point 2: Line 188: Technical implementation details should rather be documented in a GitHub readme file. 

Response 2:  We decided to leave the implementation details in the article because the two described functions fitscdiscr and predict are used in Fig. 3. We believe that this is important information for researchers who would like to repeat the experiments and should remain in the article. 

Point 3: Section 2.4: IMPORTANT. First, Figure 3 shows that input to RDA is the action vector [w_1,...,w_n,delta,gamma]. It is clear how gamma is used. However, it is not clear how you use the remaining w_1,...,w_n and delta. Second, how does the classifier compute the class label? Here I need a function that takes as input the trained classifier parameters (I guess sigma?) and the gait sequence, and outputs the predicted class label. 

Response 3:  We have added the description of the parameter delta and the observation weights w1,...,wn  in section 2.4. Moreover, the method of computing the classifier output is provided in (3). 

Point 4: Equation 3: The variables pbest and gbest are defined but never used in the rest of the paper. As they represent an iteratively improving solution, it is intuitive to me that they play a role in deriving the classifier parameters. Can you please put them in the corresponding equations? 

Response 4:  The variables pbest and gbest are used in the description of the PSO in equation 6. They mean the personal best and the global best positions. These are parameters specific to the PSO algorithm, but other swarming algorithms may have parameters that play a similar role. For example, in the GWO algorithm we have αβ and δ wolves, which also store the best solutions in the population. Section 2.8 (Integration of SI with RDA) describes the stages that are characteristic for all swarm algorithms, such as: initialization, evaluating the objective function or updating of agents. However, we do not describe here the update strategies used by the individual algorithms such as particles update based on pbest and gbest positions in PSO. 

Point 5: Response 4: I see it now. Then maybe I suggest two slight changes in the Figure 2: [1] Place RDA away from the swarm intelligence box, but still inside the model building box, and add an arrow from it to the swarm intelligence box; [2] Include training set and validation set inside the model building box? But if I am wrong then please disregard this. 

Response 5:  Your suggestions are good, but we believe, that our version of the drawing better reflects the idea of our algorithm. Please see figure 3, where the RDA is “inside” swarm intelligence. One of the steps of the algorithm is the block “Evaluate objective function” in which the RDA is executed.  

Point 6: Response 6: IMPORTANT. This is not what I meant. Between lines 122--123 you write that “Model training is carried out using one of the hybrid method, in which swarm intelligence techniques optimize the observation weights and hyperparameters of the RDA.” -- and it is substantial that you give the exact function to calculate the agent vector from intermediate positions (outputs) of the swarm techniques. 

Response 6: The following sentences explaining this issue have been added in section 2.8: 

“The swarm optimization algorithm generates many hypothetical solutions (which are represented by agents) and the best solution is selected in the optimization process. The algorithm returns the optimal parameters of the RDA classifier in the best agent vector. This result is included in the block 'Return the global best solution' in figure 3. On this basis, the objective function value for the test data is calculated.”